# Engineering Synthetic Microbial Communities through a Selective Biofilm Cultivation Device for the Production of Fermented Beverages

**DOI:** 10.3390/microorganisms7070206

**Published:** 2019-07-20

**Authors:** Sokny Ly, F. Bajoul Kakahi, Hasika Mith, Chanvorleak Phat, Barbara Fifani, Tierry Kenne, Marie-Laure Fauconnier, Frank Delvigne

**Affiliations:** 1Terra Research and Teaching Centre, Microbial Processes and Interactions, Gembloux Agro-Bio Tech, University of Liège, Gembloux Agro-Bio Tech, 5030 Gembloux, Belgium; 2Faculty of Chemical and Food Engineering, Institute of Technology of Cambodia, Phnom Penh 12156, Cambodia; 3General and Organic Chemistry, Gembloux Agro-BioTech, University of Liège, 5030 Gembloux, Belgium

**Keywords:** microbial interactions, volatolomic, biofilm, alcoholic fermentation

## Abstract

Production of Cambodian rice wine involves complex microbial consortia. Indeed, previous studies focused on traditional microbial starters used for this product revealed that three microbial strains with complementary metabolic activities are required for an effective fermentation, i.e., filamentous fungi (*Rhizopus oryzae),* yeast (*Saccharomyces*
*cerevisiae*), and lactic acid bacteria (*Lactobacillus*
*plantarum*). Modulating the ratio between these three key players led to significant differences, not only in terms of ethanol and organic acid production, but also on the profile of volatile compounds, in comparison with natural communities. However, we observed that using an equal ratio of spores/cells of the three microbial strains during inoculation led to flavor profile and ethanol yield close to that obtained through the use of natural communities. Compartmentalization of metabolic tasks through the use of a biofilm cultivation device allows further improvement of the whole fermentation process, notably by increasing the amount of key components of the aroma profile of the fermented beverage (i.e., mainly phenylethyl alcohol, isobutyl alcohol, isoamyl alcohol, and 2-methyl-butanol) and reducing the amount of off-flavor compounds. This study is a step forward in our understanding of interkingdom microbial interactions with strong application potential in food biotechnology.

## 1. Introduction

Traditional fermented foods have an important place in the food culture of human society worldwide, as the fermentation enhances the shelf life, texture, taste, aroma, and nutritional value of the foods [1]. In Cambodia, several fermented products have been developed from rice grain, such as rice vinegar, fermented rice (*Tampè*), and rice wine (*Sra sor*). Among those, rice wine is the most common application of rice fermentation, while it is still produced under traditional practices. Rice wine producers regularly face the problems of low yield and inconsistent quality in terms of taste and flavor. These traditional processes lack research and optimization in the field of food biotechnology. The principle of rice wine production consists of saccharification of the steamed, starchy source by fungi under solid state fermentation and alcoholic fermentation by yeasts under submerged fermentation [2,3,4]. The nature of microbial communities in Cambodian traditional starters, their interactions, and their contributions to the synthesis of aromas during fermentation are still widely unknown. Microbial diversities in fermentation starters have been studied by many researchers [5,6,7,8,9,10,11,12,13]. However, there is a very limited number of studies exploiting the interaction between these three groups of microorganisms and their effect on rice wine quality during the fermentation process. Liu et al. [14] pointed out that wine fermentation is not a single-species process, and the role of the different microbial wine-related species in wine production is the attention of worldwide research. Volatile compounds of wine produced by mixed cultures of *Saccharomyces cerevisiae* and non-Saccharomyces strains were significantly different from those made by mono-culture [15]. This indicates the important metabolic interaction between yeast strains during fermentation. As mentioned above, not only yeast but also filamentous fungi and lactic acid bacteria (LAB) are involved in rice wine production, and it would be very informative and helpful to investigate the impact of these three groups on volatile compounds produced in rice wine.

Filamentous fungi in submerged and semi-solid conditions in rice wine production can lead to either an increase of broth viscosity or a decrease of nutrient diffusion rate due to their different morphologies, ranging from dispersed filaments to pellets [16]. Large-scale applications are limited due to the appearance of oxygen and nutrient gradients inside the solid mass [17]. Optimal management of microbial communities can be achieved through the design of cultivation media promoting metabolic interactions [18] or by the design of an alternative cultivation device enhancing the spatial organization of the community [19]. The design of the biofilm cultivation device has been previously optimized to improve the natural binding of fungal biomass on inert surfaces (i.e., metal wire gauze packing). This alternative cultivation device was developed for the production and purification of hydrophobin HFBII from filamentous fungi *Trichoderma reesei* [20] and for the production of recombinant glucoamylase by *Aspergillus oryzae* [17]. In both cases, the fungal system displayed strong attachment on the metal packing, without any significant growth in the liquid phase. In the opposite case, yeast [21] and bacterial [22,23] systems exhibited lower attachment in similar devices with a significant proliferation in the liquid phase. Thus, based on the differential attachment on the inert surface, the biofilm cultivation device could be used to promote structuration of microbial species within communities involving fungi, yeast, and bacteria.

## 2. Materials and Methods

### 2.1. Strain and Medium Preparation

Yeast strain *Saccharomyces cerevisiae (Sc)*, filamentous fungi *Rhizopus oryzae (Ro)*, and lactic acid bacteria *Lactobacillus plantarum (Lp)* were isolated from Cambodian traditional dried ferment starter (*Dombea*) and used in this study. These three strains were isolated at different stages of the rice wine fermentation process. *R. oryzae* was isolated after inoculation of cooked rice with fermentation starter (saccharification step), while *S. cerevisiae* and *L. plantarum* were isolated during alcoholic fermentation. *R. oryzae* was incubated in Dichlorane Rose Bengal Chloramphenicol medium and grown for 72 h, followed by harvesting of spores and storage in 30% glycerol. *S. cerevisiae* and *L. plantarum* were inoculated in yeast extract peptone dextrose and De Man, Rogosa and Sharpe (MRS) broth, respectively, for 24 h and stored in 30% glycerol at −80 °C for further use. Pigmented rice (Red rice) and artificial liquid rice media were used and studied.

### 2.2. Fermentation Based on Red Rice

A laboratory-scale fermentation of red rice was adapted to mimic the traditional process used by local rice wine producers. Briefly, 100 g of red rice were soaked in distilled water for 3 h. The soaking water was discarded and a volume of 100mL of distilled water was then added and steamed in an autoclave at 120 °C for 20 min. The gelatinized rice paste was cooled down to room temperature and further inoculated and mixed with 2% of traditional dried starter (purchased from local producer) before being incubated at 30 °C. After solid-state aerobic fungal fermentation for 3 days, an additional volume of 100mL of sterilized water was added to boost the alcoholic fermentation for 7 more days in the same flask. The fermented rice mashes were homogenized and the sampling was performed every 24 h for 10 days.

### 2.3. Fermentation Based on Synthetic Liquid Medium

In order to investigate the interaction between these three potential culture strains, artificial liquid rice media were created based on major compounds in rice [24] and used in a packing system flask. Metal packing (wire gauze, 316 L stainless steel) used in the study by Zune and his team were adapted in order to fit in the middle of a 250 mL Erlenmeyer flask with 100 mL of media (see Figure 4B for a scheme of the device) [25]. Additionally, flasks without metal packing were considered as the control, i.e., considered to be equivalent to traditional submerged fermentation. The composition of artificial rice media was created according to the potential rice components. The mixture of soluble starch 20g/L, arginine 0.196g/L, alanine 0.151 g/L, leucine 0.214 g/L, valine 0.151 g/L, phenylalanine 0.133 g/L, glutamic acid 0.526 g/L, aspartic acid 0.242 g/L, CaCl2 0.124 g/L, FeSO_4_ 0.0012 g/L, MgCl2 0.0028 g/L, CuSO4 0.0024 g/L, MnSO4 0.0051 g/L, K2HPO4 1.584 g/L, and KH2PO4 0.58 g/L was adjusted to pH 6.5 with potassium phosphate buffer. All inocula were added into the liquid phase and started at the appropriate concentration of spores or CFU/mL.

### 2.4. Aromatic Compounds Analysis byHS-SPME-GC-MS

Analysis of the volatile compounds was performed based on Head-Space-Solid-Phase Microextraction (HS-SPME) followed by Gas-Chromatography-Mass-Spectrophotometry (GC-MS) analysis. The sample was extracted using a 50/30 µm Divinylbenzene/Carboxen/ Polydimethylsiloxane (DVB/CAR/PDMS) fiber (Supelco, Inc., Bellefonte, PA, USA). Each liquid sample (5 mL) was placed in a 20 mL SPME glass vial together with 30% *w*/*v* of sodium chloride and 1 µL of the internal standard 2-octanol (0.4095 mg/L in absolute methanol). The vial was tightly capped, shaken, and left to equilibrate for 30 min at 60 °C, and then fiber was exposed to the headspace for 30 min. The fiber was introduced into the injection port of the GC-MS system (at 250 °C for 10 min) and the analysts extracted from the fiber were thermally desorbed. The analysis was done in the splitless mode using helium at a total flow rate of 50 mL/min. The identification of the extracted compounds was performed in a Shimadzu GC-2010plus with a Rtx-5MS capillary column. The column carrier gas was helium at a flow rate of 1.5 mL/min. The mass detector operated in the electron impact mode was relative to the tuning result in a range from 35 to 550 m/z, and the ion source temperature was set at 230 °C. The oven temperature was held at 35 °C for 1min, raised at 6 °C /min to 155 °C, then raised to 250 °C at a rate of 10 °C / min, and held at 250 °C for 20min. The aromatic components were identified by comparison of their Retention Indices with data reported in the literature and their mass spectra the National Institute of Standard and Technology (NIST) 11 data base (matching quality higher than 90%). The Retention Indices (RI) of unknown compound were calculated by the retention time of a series of alkanes (C5-C35). Semi-quantitative analysis of the volatile compounds was performed using octan-2-ol as the internal standard. The results were reported on the basis of a mean value from two biological and analytical replicates.

### 2.5. Sugar and Ethanol Analysis by HPLC 

The concentrations of glucose, ethanol, and acetic and lactic acids were determined using High-Performance-Liquid-Chromatography coupled with a Refractive index detector (RID-HPLC, Shimadzu LC20A, Japan). A volume of 5 µL of the sample was injected, in duplicate, through a RezexROA-Organic Acid column (300 × 7.8 mm) with 5 mM H_2_SO_4_ as the mobile phase at a flow rate of 0.6 mL/min at 60 °C.

### 2.6. Microbiological Analysis 

Yeast and LAB growth were followed by selective plate count with dechlorane rose bengal with 0.01% chloramphenicol (Merck, Germany) and MRS agar with 0.01% cycloheximide (Merck), respectively. Biomass attached on packing was measured after drying for 24 h at 105 °C followed by subtraction of the mass of metal packing.

### 2.7. Statistical Aanalysis

ANOVA of chemical and volatile compound analysis was done for the different fermentation treatments. Mean rating and Least Significant Differences (LSD) for each treatment were calculated from each analysis of variance with Minitab 18. Principal component analysis (PCA) was performed to establish the relations of aroma compounds between samples.

## 3. Results

### 3.1. Comparison of Natural and Synthetic Community for Red Rice Wine Fermentation 

#### 3.1.1. Ethanol and Organic Acid Production during Red Rice Wine Processing

Previous studies reported that Cambodian traditional fermentation starter (natural community) contains *Rhizopus oryzae* as the main filamentous fungi, *Saccharomyces cerevisiae* as the main fermenting yeast, and *Lactobacillus plantarum* as the main lactic acid bacteria. Additionally, the presence and co-occurrence of these three microbial strains were associated with the generation of major flavor compounds. In this study, red rice wine fermentation kinetics were investigated for both culture with a natural community and synthetic community made of *R. oryzae*, *S. cerevisiae* and *L. plantarum*. Sugar, ethanol, and organic acids play important roles in wine taste and quality; accordingly, these parameters were investigated during the process (Figure 1). The time evolution of ethanol and organic acid production was correlated with the consumption of glucose. Maltose and glucose were the main reducing sugars detected in this study (Appendix A). Surprisingly, synthetic communities comprising high yeast inoculant (i.e., *Ro-Sc* 1:10 and *Ro-Sc-Lp* 1:10:1) did not lead to the highest ethanol production. This type of observation has been made previously for Chinese fermented beverages [26]. According to Table 1, the significant highest ethanol productions were observed when using the natural community, followed by the community *Ro-Sc-Lp* 1:1:1. Whereas the natural community gave the highest ethanol yield, it also led to the accumulation of acetic and lactic acid levels, two compounds considered as off-flavor compounds for wine.

#### 3.1.2. Comparative Analysis of the Impact of Natural and Synthetic Communities on the Flavor Profile during Red Rice Wine Fermentation

Flavor and aroma profiles are important factors responsible for the organoleptic quality of wine. In this study, flavors compounds produced during the fermentation process were analyzed to investigate and evaluate the efficiency of natural and synthetic communities. Thirty-nine volatile compounds, including alcohol, ester, acid, aldehyde, and ketone, were identified in rice wine mash by HS-SPME-GCMS (Table 2). The use of *L. plantarum* in the brewing process resulted in increased production of the following flavor compounds: isoamyl acetate, phenethyl acetate, ethyl octanoate, ethyl lactate, ethyl acetate, and isoamyl alcohol. These flavor compounds were considered as important aromas for wine quality, since they were recognized as being fruity and whisky-like aromas with lower detection thresholds [27]. Moreover, acetic acid was produced less by the synthetic communities compared to natural communities (Table 2 and Figure 1).

Because each rice wine sample contained numerous flavor compounds, for a better visualization and interpretation of the data, principal component analysis (PCA) was performed to identify correlation and similarity between samples. Figure 2A shows the corresponding factor loading plots establishing the relative importance of each flavor compound found in each sample. The discrimination of rice wine made by synthetic and natural communities is shown in Figure 2B. Flavor profile and ethanol yield produced by mixtures of those three groups of microorganisms in the same ratio (*Ro-Sc-Lp* 1:1:1) were similar to those of natural communities. However, the aroma profile produced by the group of *Ro-Sc-Lp* 1:10:1 was distinctly different from that of the natural community. The presence of *L. plantarum* in the synthetic community significantly influenced the aromatic profile of rice wine.

### 3.2. Spatial Structuration of Synthetic Communities Based on Biofilm Ccultivation.

#### 3.2.1. Impact of Submerged and Biofilm Cultivation on Colonization Efficiency

Most oenologists are interested in new fermentation technologies for optimizing the wine production process, either for quality or displaying particular flavor profiles [28]. However, this kind of study on rice wine production technology is very limited. In the context of mixed cultivation, especially with the presence of filamentous fungi involved in the process, biofilm cultivation design with metal structured packing is interesting in order to understand its impact on biofilm formation and microbial interactions for target flavor compounds contributing to rice wine quality. Figure 3 represents the biomass and microbial evolution on the packing and in planktonic phase during single- and co-cultures. Based on observation, the maximal biofilm development always occurred at the level of the air/liquid interface. According to Figure 3A, *S. cerevisiae* and *L. plantarum* could grow on artificial rice media containing soluble starch as the carbon source. It was also found that in liquid phase, *S. cerevisiae* and *L. plantarum* promoted the growth of each other in a mutualistic way (Figure 3A,C–E) by comparison with single culture. The presence of *L. plantarum* in all cases did not affect either ethanol production or biomass attachment on packing. Ethanol production was observed in significant amounts for the cultures made with *S. cerevisiae*, single-, or co-culture (Figure 3B). When *S. cerevisiae* is absent, only very low amount of ethanol is observed. In the same way, the presence of *S. cerevisiae* in co-culture contributed to higher amount of biomass when a biofilm cultivation device was used (Figure 3C). On the other hand, there was no evolution for either *S. cerevisiae* or *L. plantarum* on the metal packing when they were grown without *R. oryzae* (data not shown), showing the role of *R. oryzae* in initial colonization.

#### 3.2.2. Impact of Biofilm Mode of Cultivation on Flavor Compound Using Natural and Synthetic Communities

Natural and synthetic communities were cultivated in biofilm mode in order to investigate the impact of species structuration on flavor production. As a result, using biofilm cultivation mode led to a significant improvement of flavor compound production in comparison with submerged culture, for both for natural and synthetic communities (Table 3). As expected, the metallic support of the biofilm cultivation device sustained the selective development of *R. oryzae*, promoting the structuration of the whole community. This mode of fungal development is also recognized as increasing secretion abilities, probably leading in our case to a higher amylolytic activity. Additionally, some unpleasant compounds, i.e., pentanoic acid, isovaleric acid, and acetic acid, were produced in high amounts during classical submerged cultivation and were absent in the biofilm mode of cultivation. This mode of cultivation also led to an increase of some flavor compounds, which are important for the organoleptic properties of fermented beverages, i.e., phenylethyl alcohol and isoamyl alcohol. 

## 4. Discussion

After isolating some microbial strains from a natural community, targeted experiments can be performed to highlight optimal community compositions that are responsible for the production of specific metabolites through specific biochemical interactions [29]. In this work, we applied this concept to isolate three strains, i.e., *R. oryzae*, *S. cerevisiae* and *L. plantarum*, from a natural community used in Cambodian fermentation starter in order to determine the performances of synthetic communities by comparison with the natural one. Indeed, microbial composition has a strong impact on both rice wine quality and yield. Basically, spontaneous cereal-based fermentation involves complex interkingdom microbial consortia, including fungi, yeasts, and lactic acid bacteria [2,13]. *R. oryzae* has been reported as a strong amylase enzyme producer, frequently found in amylolytic fermentation starters for rice wine and during saccharification [4,7,9,30,31]. *R. oryzae* produces amylase hydrolyzing starch into fermentable sugar feeding *S. cerevisiae* and *L. plantarum* (Figure 4A). The presence of LAB found in cereal fermentation is important because beside producing lactic acid, LAB is likely to contribute to the production of some flavor compounds and to display some specific metabolic cross-feeding with yeast [32]. The growth of *S. cerevisiae* in fermented food is induced by acidification from bacteria and it adjusts its metabolism by secreting a serial metabolite, notably an amino acid, allowing the survival of LAB (Figure 4A) [33,34]. The result of this study supports this evidence, with both *L. plantarum* and *S. cerevisiae* promoting the growth of each other during the first 24 hours when grown in co-culture.

The major flavor compounds found in this study were ester group that provide a pleasant flavor to wine. The type of ester formed depended on the fermentation environment including temperature and level of alcohol [35]. Guitart and his team revealed that high concentrations of amino acids in grape must was also shown to enhance the production of more volatile esters [36]. Similarly, red rice is considered as rice containing highest protein as well as free amino acid by comparison with white sticky and non-sticky rice [37]. Beside this, microbial interaction also has a tremendous impact on flavor compound secretion. Based on Table 2, the type and concentration of ester group, as well as other groups, were affected by the combination of species. Metabolite production can indeed be considerably modified by the microbial composition [14,38]. Based on the results accumulated during this work, we have shown that a simple co-culture *Ro-Sc-Lp* 1:1:1 could provide high ethanol yield and alike flavor profile by comparing to the natural community; moreover, it could reduce some undesired flavors and lower acid which could have off-flavor and taste, especially acetic acid. This might probably because of various and unknown microorganism involved in traditional brewing process and led to difficult control and microbial competitive interaction [13]. The different combination of these three strains has somehow different impacts on chemical compositions of rice wine including lactic acid, acetic acid, and ethanol yield. These three strains together were able to produce flavor profile which was similar to that from natural microbial community, regularly used in rice wine production in Cambodia. 

Beside the optimization of the cultivation medium and consortia members, alternative cultivation device can also be proposed. The selection of microbes for specific aroma profile cannot be effective without understanding how microbe interact with each other [14]. In such a case, a biofilm cultivation has been used for promoting the spatial structuration of the three microbial strains extracted from the original starter [17,23]. Indeed, the metallic support used in this device have been previously reported as enhancer the natural binding of filamentous fungi *Trichoderma reesei* [20] and *Aspergillus oryzae* [17]. The design of cultivation media promoting metabolic interaction and the design of alternative cultivation device enhancing natural spatial organization of community are both necessary for the study of microbial community [18,19].

Fungal spores generally prefer to adhere and develop on lateral surfaces by several mechanisms involving complex interactions between physical and biological factors. Thus, inserting solid support in the middle flask is very important for the growth of filamentous fungi. Filamentous fungi in submerged and semi-solid conditions always cause problem of increasing viscosity and forming different morphologies leading difficulties of system control during production, notably; cereal-based fermentation [16]. Rice wine production techniques were commonly performed in simultaneous saccharification and fermentation (SSF) method by which starch enzymatic hydrolysis and alcoholic fermentation occur simultaneously in the same reactor. Using SSF can eliminate the inhibition of saccharifying enzyme by sugar substrate [39] and give unique flavor of rice wine [4,40]. The process of rice wine production operating with enzymatic saccharification and alcoholic fermentation gave distinguish flavor profile. This is why saccharification by fungi, alcoholic fermentation by yeast are still used for rice wine production practice. Recently, novel functionality was achieved within a bacterial–fungal co-culture to create special flavors during fermentation in the food industry [41]. Microbial interactions are very essential for a successful establishment and maintenance of a microbial population [42]. Synthetic community ecology focuses on designing, building and analyzing the dynamic performance and understanding how community properties appear as a consequence of those interactions [43]. In the field of biotechnological applications, single- and multi-species culture have a rise of interest in field of research in order to investigate new technology as well as interested metabolites [44,45]. The mixture of chemical substance according to rice composition has been used to set up a new technology using corrugated metal in submerge fermentation flask. The support can be considered as an efficient way for bioprocess intensification by promoting the exchanges between the biofilm, gas and liquid phases. The result revealed the importance of metal packing system on the filamentous fungi growth and interaction with other strains during the process. The biomass binding on packing, the growth rate and ethanol production in planktonic were affected by the combination of strains and the presence of solid support. Moreover, the microbial interaction in term of flavor production was really occurred. Even though *R. oryzae, S. cerevisiae* and *L. plantarum* could grow alone in this media, single species was not able to produce some compounds; however, co-culture of three strains could secrete some important compounds such as phenylethyl alcohol, isobutyl alcohol, isoamyl alcohol and 2-methyl-butanol. Additionally, some of off-flavor compounds (pentanoic acid, isovaleric and acetic acid) were reduced in biofilm cultivation mode. 

Cambodian people have their own preference for the typical sweet-sour-floral aroma of rice wine. However, keeping or replacing this particular product is a key role for researchers to figure out a synthetic community to produce rice wine in a better control system with similar organoleptic and consistent qualities. Consequently, the information from this study can be a part of an improvement of Cambodian traditional dried starters and provide insight and understanding into traditional rice wine development. Moreover, the biofilm cultivation device from this study is a tool that enables a step forward in developing and improving the understanding of interkingdom microbial interactions involving filamentous fungi, yeast, and bacteria in the aspect of optimal management of organization of microbial communities.

## Figures and Tables

**Figure 1 microorganisms-07-00206-f001:**
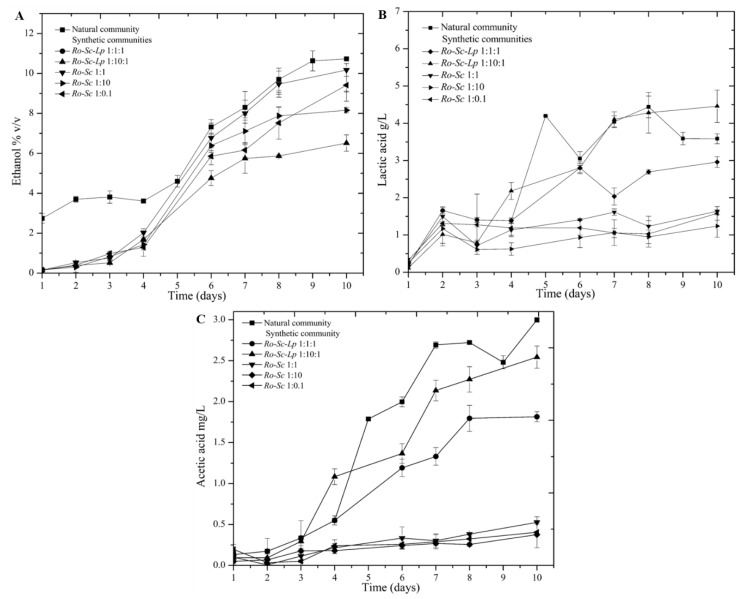
Kinetics of ethanol (**A**), lactic acid (**B**)**,** and acetic acid (**C**) during fermentation by natural and synthetic communities. Synthetic communities were prepared with different ratios of microbial species. In all cases, *Rhizopus oryzae* was inoculated at an initial concentration of 10^6^ spores/ml. For the cultures involving either *S. cerevisiae* or *L. plantarum*, their initial concentrations are indicated by 0.1, 1, or 10, corresponding to 10^5^, 10^6^, and 10^7^ CFU/mL, respectively.

**Figure 2 microorganisms-07-00206-f002:**
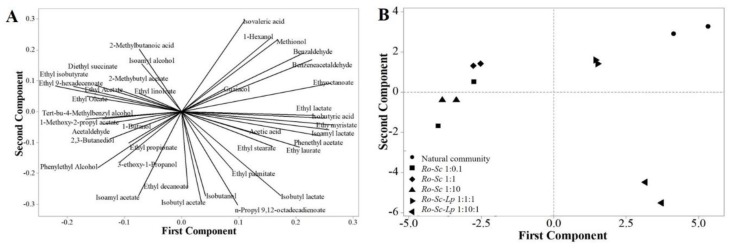
PCA) plot displaying the different aromatic profiles of rice wine by natural and synthetic communities. (**A**) plot of line distribution of 39 volatile compounds. (**B**) PCA plot discriminating each sample. In all cases, *R. oryzae* was inoculated at an initial concentration of 10^6^ spores/ml. For the cultures involving either *S. cerevisiae* or *L. plantarum*, their initial concentration is indicated by 0.1, 1, or 10, corresponding to 10^5^, 10^6^ and 10^7^ CFU/mL, respectively. The same symbols represent the samples from biological replications.

**Figure 3 microorganisms-07-00206-f003:**
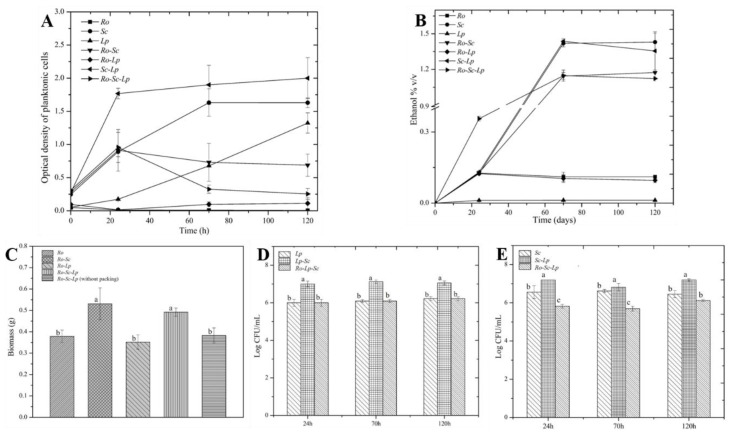
Biofilm and microbial evolution during single and co-culture in biofilm reactor. (**A**) Optical density OD of the planktonic phase. (**B**) Ethanol production. (**C**) Total biomass of the biofilm attached on packing and without packing after 70 h. (**D**) L. plantarum and (**E**) S. cerevisiae plate count while in the planktonic phase. Means with the same letter are not significantly different from each other (p < 0.05).

**Figure 4 microorganisms-07-00206-f004:**
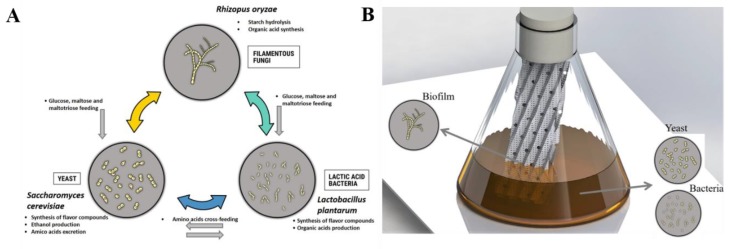
Scheme showing microbial interaction in biofilm cultivation device. (**A**) hypothetical metabolic interactions between fungi, yeast, and LAB, and the resulting functions during red rice wine fermentation (or on the synthetic liquid medium). (**B**) Scheme showing the biofilm cultivation device used in this study, i.e., a shake flask with a stainless steel packing sheet, as well as the expected positions of fungi, yeast, and LAB species during fermentation.

**Table 1 microorganisms-07-00206-t001:** Final concentration of ethanol and organic acids at day 10, the end of the traditional rice wine brewing process.

Concentration	Natural Community	*Ro-Sc-Lp* 1:1:1	*Ro-Sc-Lp* 1:10:1	*Ro-Sc* 1:1	*Ro-Sc* 1:10	*Ro-Sc* 1:0.1
Ethanol (% *v*/*v*)	10.73 ± 0.07 ^a^	9.95 ± 0.02 ^b^	6.5 ± 0.41 ^d^	10.17 ± 0.32 ^b^	7.95 ± 0.13 ^c^	8.41 ± 0.80 ^c^
Acetic acid (g/L)	2.99 ± 0.02 ^a^	1.81 ± 0.06 ^c^	2.54 ± 0.13 ^b^	0.52 ± 0.02 ^d^	0.37 ± 0.01 ^d^	0.40 ± 0.18 ^d^
Lactic acid (g/L)	3.58 ± 0.13 ^a^	2.95 ± 0.14 ^b^	4.45 ± 0.43 ^a^	1.64 ± 0.01 ^c^	1.23 ± 0.30 ^c^	1.58 ± 0.20 ^c^

Values in a row that are not preceded by the same letter are significantly different (*p* ≤ 0.05).

**Table 2 microorganisms-07-00206-t002:** Flavor compounds (μg/L) produced from the natural community and a combination of strains as the synthetic communities in the red rice wine brewing process.

N	Compounds	RI	Natural Community		Synthetic Community													
					*Ro-Sc-Lp* 1:1:1			*Ro-Sc-Lp* 1:10:1			*Ro-Sc* 1:1			*Ro-Sc* 1:10			*Ro-Sc* 1:0.1	
			Average	STD	Average	STD	Average		STD	Average		STD	Average		STD	Average		STD
**Ester**			**19506.02**		24029.27		21738.23			28584.87			33152.54			19290.85		
1	2-Methybutyl acetate	880	Nd		137.89	15.85	Nd			55.07		21.45	170.05		4.09	Nd		
2	Isoamyl acetate	876	125.98	15.4	270.1	107.15	493.09		123.93	155.8		9.3	533.14		4.66	344.24		193.83
3	1-Methoxy-2-propyl acetate		Nd		Nd		Nd			Nd			157.95		33.39	18.83		
4	Ethyl linoleate		60.44	36.9	41.39	9.75	Nd			106.44			213.92		195.68	Nd		
5	4-Methylbenzyl alcohol		Nd		Nd		Nd			30.7			35.16			35.42		
6	Phenylethyl acetate	1252	1383		1421.55	346.22	1718.85		40.95	427.9		371.34	683.62		71.9	98.05		62.35
7	Ethyl isobutyrate	756	Nd		275.28	61.82	Nd			270.53		149.22	296.72		74.47	381.78		175.2
8	Ethyl propionate	713	Nd		131.79	55.24	172.86		100.09	186.71		63.47	416.39		294.43	101.98		10.34
9	Ethyl myristate	1651	1206.16	63.67	1202.58	70.77	1296.14		123.41	80.21		113.44	216.05		56.89	61.84		31.72
10	Ethyl octanoate	1041	294.15	5.68	134.73	18.25	154.17		7.44	38.33		54.21	41.68		1.137	20.75		
11	Isobutyl lactate		51.88	48.02	43.82	17.34	266.86		14.88	8.96		12.68	29.6			Nd		
12	Ethyl lactate	1010	4436.35	1501	3781.81	1016.62	3941.33		635.91	941.16		479.84	2313.94		1345.6	1137.4		
13	Propyl 9,12-octadecadienoate		Nd		17.65	7.71	59.41		15.7	Nd			Nd			20.92		6.88
14	Ethyl stearate		253.6	154.03	121.71	37.42	366.97		1.31	234.15		70.35	179.59		68.49	20.74		
15	Ethyl decanoate	1398	63	1.42	65.25	7.81	269.24		204.4	160.2		106.35	154.6		3.35	90.31		18.23
16	Ethyl laurate	1494	281.26	53.27	213.47	23.88	412.84		1.29	241		144.21	109.99		1.69	44.13		18.64
17	Ethyl 9-hexadecenoate		Nd		Nd		Nd			122.79		64.19	61.07		86.37	24.47		12.34
18	Ethyl Acetate	628	9205.5	1011.45	14189.71	4590.13	4593.61		391.19	20976.76		8697.86	24587.57		7650.87	15713.3		0
19	Ethyl Oleate		189.26	58.42	196.77	86.69	148.4		28.87	594.82		164.86	669.91		776.57	103.1		53.28
20	Ethyl palmitate		1765.79	162.58	1467.66	213.54	7218.18		883.39	3420.41		1023.08	1901.87		129.36	537.84		343.77
21	Isoamyl lactate		149.34	14.15	34.34	14.56	134.23		15.56	Nd			32.67		23.1	Nd		
22	Diethyl succinate	1167	Nd		5.43	1.6	Nd			28.61		14.22	22.1		0.78	12.27		
23	Isobutyl acetate	776	Nd		214.98	105.78	388.61		13.36	87.83		2.41	242.27		38.11	57.66		45.7
**Acids and Aldehyde**			7525.45		5551.64		7215.71			1274.4			8219.13			588.64		
24	2-Methylbutanoic acid		254.71	28.66	158.23	56.37	Nd			204.84		76.25	352.89		67.68	79.54		82.67
25	Isovaleric acid	877	157.19	47.88	108.41	18.89	Nd			125.98		15.88	22.24			22.19		15.69
26	Acetic acid	600	6050.36	225.56	4602.05	75.1	6487.79		1225.43	85.13		111.22	7576.43		524.84	Nd		
27	Isobutyric acid	1215	677.61	82.74	536.91	83.6	632.95		16.75	420.74			205.59		75.61	120.04		138.59
28	Acetaldehyde		Nd		6.07	5.5	39.12		1.15	387.05			Nd			366.87		219.63
29	Benzaldehyde	960	134.63	14.02	40.06	7.93	22.17		14.06	16.61			23.25			Nd		
30	Benzeneacetaldehyde		250.92	23.21	105.95	38.56	72.78		8.01	34.05		6.04	38.72		6.18	Nd		
**Alcohol**			46685.3		47489.59		47434.43			56613.62			68742.36			51864.3		
31	1-Hexanol	851	108.9	10.82	137.59	4.57	33.29		1.49	62.15		7.74	14.8		7.34	33.65		47.6
32	3-ethoxy-Propanol	833	40.23	2.67	55.19	34.44	64.25		17.715	29.33		7.05	82.57		12.27	98.85		52.61
33	Isobutanol	647	9854.79	262.93	8041.47	2.28	22432.98		1803.43	17018.02			12945.72		2849.94	12593.74		1040.56
34	Methionol	978	266.39	142.01	266.71	30.07	Nd			42.58			64.56			5.93		8.4
35	Isoamyl alcohol		31276.83	2546.51	30257.95	536.34	21182.44		1471.7	34607.6		11519.76	34191.78		4674.47	24232.3		12570.1
36	Guaiacol	1089	131.53	4.58	85.42	13.26	93.52		12.09	115.24		44.51	125.01		28.39	49.07		
37	Phenylethyl Alcohol	1118	3294.85	534.53	3446.39	576.43	8502.17		1331.22	9527.54		2295.83	9903.82		4338.51	6696.66		3835.75
38	1-Butanol	675	167.66	28.22	530.19	92.92	343.05		311.04	294.54		81.73	895.69			507.3		143.73
39	2,3-Butanediol	806	1584.31	460.75	4668.68	736.44	4026.13		744.14	3648.25		755.42	10600.96		6533.65	7745.62		
**Total**			73716.77		77015.35		85567.57			94788.2			110114.03			71376.9		

STD: Standard deviation. Nd: Not detected.

**Table 3 microorganisms-07-00206-t003:** Flavor compounds (μg/L) produced from natural community and combinations of strains as synthetic communities in a biofilm cultivation device with artificial rice media.

Compounds	Natural Community		Synthetic Community *Ro-Sc-Lp*	
	With Packing	Without Packing	With Packing	Without Packing
Isoamyl acetate	1.78 ± 0.3	2.00 ± 0.46	Nd	Nd
Phenethyl acetate	1159.08 ± 69.74	7.92 ± 0.95	Nd	Nd
Ethyl propionate	4.85 ± 0.15	Nd	Nd	Nd
Ethyl myristate	3.29 ± 0.1	Nd	Nd	Nd
Ethyl stearate	3.03 ± 0.8	Nd	Nd	Nd
Ethyl laurate	Nd	26.58 ± 16.80	Nd	Nd
Ethyl Acetate	3.16 ± 1.19	1.65 ± 0.12	Nd	Nd
Ethyl lactate	Nd	Nd	2.5 ± 0.2	Nd
Ethyl Oleate	9.12 ± 0.98	2.97±1.61	Nd	Nd
Ethyl Isobutyrate	Nd	Nd	16.62 ± 0.89	Nd
Ethyl palmitate	28.33 ± 17.45	3.64 ± 1.07	Nd	Nd
Isobutyl acetate	43.14 ± 6.43	34.01 ± 14.05	Nd	Nd
Isovaleric acid	Nd	23.69 ± 6.19	Nd	275.58 ± 45.67
Pentanoic acid	Nd	Nd	124.72 ± 13.45	593.58 ± 34.23
Acetic acid	23.31 ± 2.63	8.53 ± 6.03	4.13 ± 0.56	1017.38 ± 21.12
1-Hexanol	40.66 ± 6.89	Nd	Nd	18.82 ± 0.93
Isobutanol	3.24 ± 0.3	210.87 ± 140.32	Nd	Nd
Methionol	Nd	4.35 ± 2.75	Nd	Nd
Propanol	Nd	Nd	10.41 ± 2.34	Nd
Butanol	Nd	Nd	10.72 ± 3.23	Nd
1-Methyl-butanol	Nd	Nd	5.04 ± 0.12	Nd
2-methyl-butanol	Nd	Nd	158.59 ± 13.89	Nd
Isobutyl alcohol	Nd	Nd	378.19 ± 16.78	Nd
Isoamyl alcohol	2374.62 ± 64.32	1703.32 ± 19.8	4247.63 ± 234.89	256.69 ± 13.89
Guaiacol	3.29 ± 0.5	Nd	Nd	3594.66 ± 84.91
Phenylethyl Alcohol	1902.86 ± 89.90	0.58 ± 0.41	2017.87 ± 89.78	Nd
2,3-Butanediol	47.80 ± 1.21	Nd	Nd	2070.68 ± 34.37
Benzaldehyde	25.50 ± 5.25	136.39 ± 17.50	91.75 ± 7.62	Nd
Acetaldehyde	2.78 ± 0.67	Nd	Nd	559.68 ± 23.34

Nd: Not detected.

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
