# Peer review of "Engineering Synthetic Microbial Communities through a Selective Biofilm Cultivation Device for the Production of Fermented Beverages"

_microorganisms, 2019, doi:10.3390/microorganisms7070206_

Round 1

Reviewer 1 Report

Nice paper on the application of a biofilm cultivation device for the production of rice wine using microbial communities that produce the alcohol and flavour.

Some comments and suggestions to improve the manuscript:

In general, grammar and spelling should be checked (if possible, by a native speaker) and sentences examined whether they really express what the authors wish to state. Some examples can be found in the listing below.

L74. The isolated strains were inoculated in ..... medium (which medium), grown for ... hours followed by harvesting cells and stored in …% glycerol.

Was Rhizopus oryzae stored as mycelium or as spores?

L92. FeSO4

L94. was adjusted pH to 6.5. With what?

L132. Contains, not “involving”

L147. Figures are too small for easy reading. The size of the graphs in supplemental data is much better. Also increase the contrast between the data points in the figures with graphs. Use other markers for the data points the various triangles are hard to distinguish.

L153. Delete “error!”

L162-163. Do you mean: “The use of L. plantarum in the brewing process resulted in (more) production of the following flavor compounds:” Rewrite this sentence.

L165. whisky-like

L166-167. Statement is partly correct because also the starters without Lp contained less acetic acid. Rewrite sentence.

L168. “and interpretation of the data”

L175-176. This statement needs some explanation. Indicate which compound and/or number in Table 2 and which data points of Figure 2 show the significantly influenced. Probably good for the general reader not that familiar with these presentations of data to explain what indicates the difference between the communities.

L177 & 224. Reduce size of Tables 2 and 3 to the compounds of interest and show all data in supplemental. Highlight data that are of importance for conclusions or show major differences. i.e. which compounds make the difference for the different communities.

L212. cultivated

L232-233. …… in order to determine the performance of the synthetic community ….

L243. What is a serial metabolite? Explain in text.

L252-253. …. compounds with ester groups that provide a pleasant flavour to wine ..

L262. What is a complementary flavor profile? Explain in the text.

L270. their own preferent habitat? Explain. Do you mean: have their own preference for the typical .....

L285. to adhere

L328. BF and BF? There is only one BF in the author list.

Author Response

In general, grammar and spelling should be checked (if possible, by a native speaker) and sentences examined whether they really express what the authors wish to state. Some examples can be found in the listing below.

 Thank you very much for this comment.  The paper has been processed by a proofreading service.

L74. The isolated strains were inoculated in .... medium (which medium), grown for ... hours followed by harvesting cells and stored in …% glycerol.

 The section 2.1 has already been modified accordingly. 

Was Rhizopus oryzae stored as mycelium or as spores?

 Rhizopus oryzae has been stored as spores in glycerol solution. 

L92. FeSO4

 It has been corrected accordingly (L101).

L94. was adjusted pH to 6.5. With what?

 pH was adjusted by phosphate buffer solution (L103)

L132. Contains, not “involving”

 It has been corrected accordingly (L147).

L147. Figures are too small for easy reading. The size of the graphs in supplemental data is much better. Also increase the contrast between the data points in the figures with graphs. Use other markers for the data points the various triangles are hard to distinguish.

 This figure has been changed accordingly. 

L153. Delete “error!”

 This has been done (L169).

L162-163. Do you mean: “The use of L. plantarum in the brewing process resulted in (more) production of the following flavor compounds:” Rewrite this sentence.

 Yes, this has been changed accordingly (L178-179). 

L165. whisky-like

 This has been changed accordingly (L182).

L166-167. Statement is partly correct because also the starters without Lp contained less acetic acid. Rewrite sentence.

 The sentence has been reformed (L182-183). 

L168. “and interpretation of the data”

 This has been corrected accordingly (L186). 

L175-176. This statement needs some explanation. Indicate which compound and/or number in Table 2 and which data points of Figure 2 show the significantly influenced. Probably good for the general reader not that familiar with these presentations of data to explain what indicates the difference between the communities.

 The paragraph has been modified and added some sentences to explain more (L185-189).

L177 & 224. Reduce size of Tables 2 and 3 to the compounds of interest and show all data in supplemental. Highlight data that are of importance for conclusions or show major differences. i.e. which compounds make the difference for the different communities.

 Thank you very much for your suggestion, it would be very much easier to understand if we highlight only the important data. However, Table 2 and 3 shown the flavor profile of each sample. It represented the volatomic profile produced by each condition and should be exhibited in a complete profile rather than just the important parts. In addition, it is very difficult to select the data to emphasize as the results that should be elaborate are not only the concentrations of detected compounds but also the number of identified compounds. For these reasons, we hope that you consider our intention. 

L212. Cultivated

 This has been changed accordingly (L231).

L232-233. …… in order to determine the performance of the synthetic community ….

 This has been changed accordingly (L251). 

L243. What is a serial metabolite? Explain in text.

 We are not able to make explanation in the text. Serial metabolite means that those metabolites were secreted subsequentially.

L252-253. …. compounds with ester groups that provide a pleasant flavour to wine.

 This has been corrected accordingly (L271). 

L262. What is a complementary flavor profile? Explain in the text.

 That word has been changed to “alike flavor profile” (L281). 

L270. their own preferent habitat? Explain. Do you mean: have their own preference for the typical .....

 The sentence has been changed accordingly (L336). 

L285. to adhere

 This has been changed accordingly (L304).

L328. BF and BF? There is only one BF in the author list.

 Thank you very much. The name of author has been changed appropriately (L353).

Reviewer 2 Report

This manuscript describes the characterization of a multi-species microbial community and comparison of this community to a naturally occurring community in terms of their ability to produce rice wine.  The concept of moving from unknown, potentially less controllable/understandable mixtures of microbes to a defined, tunable mixture of known microbes is promising in that it will allow for future understanding and refinement of the fermentation process. 

Comments

1.       All of this work is based on the initial estimated ratios of the strains.  However, these ratios likely change over time and the ratio at any given timepoint may not be the same as the starting ratio.  It would be very useful to have some analysis of the final ratio of the 3 strains in the synthetic community.  The final ratio will shed light on what is actually happening as the strains interact with one another.

2.       It might be helpful to have a schematic describing where certain samples were collected and how they were analyzed (gas phase, surface attached, supernatant, cells in suspension, etc) to clarify the overall experimental approach.

3.       Please carefully review the English throughout –there are some grammar mistakes in the document. 

4.       Line 51 – please define LAB.

5.       Line 52 – “it would be interesting” – hopefully even more than interesting but informative and helpful to build up a body of knowledge.

6.       Line 55 – refers to a diffusion rate – diffusion of what?

7.       Section 2.1 – do the strains have names?  Is there a reference for the strain isolation?  Line 231 suggests that the strains were isolated within this work, but the isolation of the stains is not discussed.  Please either include how the strains were isolated, or provide a reference that describes that process.

8.       Line 78 – was the soaking water discarded?

9.       Line 81 – where did the “traditional dried starter” come from?  For the aerobic fermentation, what was the temperature?  Was it a closed container or open to ambient air?  Was it mixed?  Please provide fermentation conditions.

10.   Line 84 – was the sampling every 24 h for 10 days?  Please add this detail.

11.   Section 2.3 does not include the details about how the artificial communities were prepared – how were the various strains mixed?  At what ratios?  This is mentioned later, but it would be helpful to mention it in the materials/methods section.

12.   Section 2.3 – how was the artificial liquid rice media sterilized?  And where did the recipe for the media come from?  Is there a reference for this formulation?

13.   Line 96-97 – define HS-SPME, GC-MS, DVB/CAR/PDMS

14.   Line 98-99 is the NaCl concentration 30% mass per volume? 

15.   Line 117 – 5 ul of what solution was injected? 

16.   Line 116 – define RID-HPLC

17.   Line 122 – in order to calculate the biomass, were the packing materials also characterized for the starting mass prior to adding them to the culture systems?  Was the biomass determined by subtracting the initial mass from the final mass?  Please include these details

18.   Methods – how many times were experiments repeated, and how many replicates were included in each experiment?  Please include this information for all datasets in the paper.

19.   Line 143 – “highest” ethanol production – is this supported by the statistics?  Please include that information.

20.   Line 153 – the “end” of the process – what timepoint was this?  10 days?  Please include.

21.   Figure 1 – the text size is quite small in the figures, and it is challenging to identify the various symbols within the charts. 

22.   Lines 166-167 – Table 1 indicates that the acetic acid production is only barely lower with the presence of Lp as compared to the natural community.  Why does this data differ from Table 2 and Fig 1?

23.   Line 184 – how many biological replicates?  Looks like perhaps 2?  In each biological replicate (I’m assuming these are on different days??) how many samples?

24.   Line 208 – did the methods describe how the biofilm was grown without packing?  And how was total biofilm mass determined in the absence of packing?

25.   Line 241 – is the metabolic cross-feeding known?  Can you point to specific compounds within your dataset to indicate that these are the metabolites of interest?

26.   Figure 4A – please be more specific.  For instance, the two-way arrows between the 3 species are a bit misleading, as some metabolites go one-way only, while others may go both ways.  Is it possible to clarify this?

27.   Fig 4B is great and would be helpful to have earlier in the manuscript when describing the approach.

28.   Line 307 – maximum biofilm at air-liquid interface sounds more like results.  Is this due to oxygen requirements?  To mention this new data in the final paragraph may not be the best approach. 

29.   Line 317 – I think the cultivation device is a tool that will help to enable steps toward developing an improved understanding of the interactions.  Please be careful of the wording in this sentence.

30.   Line 318-319 indicates “management of spatial organization” – I don’t see this being accomplished in the current work.  While there was a substrate for surface-associated communities to form, this wasn’t quite spatial organization.

31.   It would be helpful to have a conclusions paragraph or section.  The final paragraph is partially conclusions but partially new information.  Conclusions should not introduce new information but simply summarize the work and its findings.

32.   Are these 3 strains sufficient to ferment rice wine?  Do more strains need to be added?

Author Response

1. All of this work is based on the initial estimated ratios of the strains.  However, these ratios likely change over time and the ratio at any given timepoint may not be the same as the starting ratio.  It would be very useful to have some analysis of the final ratio of the 3 strains in the synthetic community.  The final ratio will shed light on what is actually happening as the strains interact with one another.

 This study has been indeed focused on the ratio between the strains at the beginning of the experiments. However, we also considered the use of selective biofilm attachment in order to promote the spatial structuration of the community, and more specially the specific attachment of the fungi (responsible of the first metabolic step, i.e. starch hydrolysis). In this case, determination of the ratio between the different strains is more tricky because of potential non selective attachment of yeast and LAB on the fungi layer. We are actually working on a RT-qPCR protocol for the determination of the ratio between strains. However, we are convinced that the results presented in this work are already relevant, notably because the volatomic profile is consistent with our initial hypotheses. This volatomic profile can indeed be used as a strong indicator of the phenotypic diversity of the synthetic community in relation with fermentation of red rice wine.

2.       It might be helpful to have a schematic describing where certain samples were collected and how they were analyzed (gas phase, surface attached, supernatant, cells in suspension, etc) to clarify the overall experimental approach.

 Thank you very much for your valuable comment. Concerning this mater, the schematic figure showing microbial interaction in biofilm cultivation device and flask with stainless steel packing sheet, as well as the expected position of fungi, yeast and LAB species during fermentation were shown in figure 4 in Discussion part. Moreover, this figure was also mentioned earlier in Material and Method part (L96 ). This figure gives enough information on experimental design. 

3. Please carefully review the English throughout –there are some grammar mistakes in the document. 

 Thank you very much for this comment.  The paper has been processed by a proofreading service.

4. Line 51 – please define LAB.

 It has been already defined in the text (L51). 

5.       Line 52 – “it would be interesting” – hopefully even more than interesting but informative and helpful to build up a body of knowledge.

 The sentence has been changed accordingly (L52). 

6.       Line 55 – refers to a diffusion rate – diffusion of what?

 This has been changed to “nutrient diffusion rate” (L55).

7.       Section 2.1 – do the strains have names?  Is there a reference for the strain isolation?  Line 231 suggests that the strains were isolated within this work, but the isolation of the stains is not discussed.  Please either include how the strains were isolated, or provide a reference that describes that process.

 Those strains were isolated from Cambodian traditional dried starter and identified by using specific primer for conservative region of bacteria (16S) and eukaryote (ITS). The detail of isolation process has been added in section 2.1. 

8.       Line 78 – was the soaking water discarded?

 Yes, the soaking water was discarded and this sentence has been added (L85). 

9.       Line 81 – where did the “traditional dried starter” come from?  For the aerobic fermentation, what was the temperature?  Was it a closed container or open to ambient air?  Was it mixed?  Please provide fermentation conditions.

 Traditional dried starter was purchased from local producer and this sentence has been added in the text. 

 The fermentation flask has been closed by cotton cover lid and incubated at 30 oC.

 Normally, the fermentation was mixed before sampling only. This was already mentioned in the text (L84-85). 

10.   Line 84 – was the sampling every 24 h for 10 days?  Please add this detail.

 Yes, this is already added in the text (L91). 

11.   Section 2.3 does not include the details about how the artificial communities were prepared – how were the various strains mixed?  At what ratios?  This is mentioned later, but it would be helpful to mention it in the materials/methods section.

 This has been added to the section 2.3. 

12.   Section 2.3 – how was the artificial liquid rice media sterilized?  And where did the recipe for the media come from?  Is there a reference for this formulation?

 The soluble starch, mineral and amino acid were mixed and sterilized. Except, glutamic acid and aspartic acid, not heat resistant, were filtered (0.45 μm). The recipe of media was made from the major compound of rice (DOI: 10.1016/j.foodchem.2010.05.115) and this source has been added in section 2.3 (L94). 

13.   Line 96-97 – define HS-SPME, GC-MS, DVB/CAR/PDMS

 Those abbreviations have been defined (L106-109).

14.   Line 98-99 is the NaCl concentration 30% mass per volume? 

 NaCl concentration was 30% w/v mass per volume, and this has been added in text(L110). 

15.   Line 117 – 5 ul of what solution was injected? 

 This has been changed to “5 μL of sample was injected” (L130).

16.   Line 116 – define RID-HPLC

 This has been defined (L129). 

17.   Line 122 – in order to calculate the biomass, were the packing materials also characterized for the starting mass prior to adding them to the culture systems?  Was the biomass determined by subtracting the initial mass from the final mass?  Please include these details

 The detail has been added to the text (L136-137).

18.   Methods – how many times were experiments repeated, and how many replicates were included in each experiment?  Please include this information for all datasets in the paper.

 The number of replications was added in line #126#. We conducted the experiment in different replication. For example, biomass quantification was done in more than 5 biological replications. However, for GC analysis, sample were collected from two biological replicates and analyzed by GC two time for each sample. 

19.   Line 143 – “highest” ethanol production – is this supported by the statistics?  Please include that information.

 Yes, this’s based on statistical analysis, refer to Table 1 (L158). 

20.   Line 153 – the “end” of the process – what timepoint was this?  10 days?  Please include.

 This timepoint was added (10 days) (L170). 

21.   Figure 1 – the text size is quite small in the figures, and it is challenging to identify the various symbols within the charts. 

 The figure 1 has been reformed to bigger size. 

22.   Lines 166-167 – Table 1 indicates that the acetic acid production is only barely lower with the presence of Lp as compared to the natural community.  Why does this data differ from Table 2 and Fig 1?

 Table 1 represented the data from HPLC analysis where Table 2 from GCMS analysis.These two different data cannot be absolutely compared with each other. The factors that made this difference are: 

- different detector: refractive index detector and Mass spectrophotometry detector.  

- different method of extraction where sample was injected directly in HPLC. Samples were extracted by SPME before being injected in GC. 

- the method SPME-GCMs was only considered as semi-quantitative analysis. It represented the relative concentration of each compounds in free head space of vial after incubated 5 mL of sample in 20 mL vial.  

23.   Line 184 – how many biological replicates?  Looks like perhaps 2?  In each biological replicate (I’m assuming these are on different days??) how many samples?

 For volatile flavor compound analysis, it was made from two biological analysis and two times of GC analysis. All fermentation was conducted at the same time and sample analysis was also done at the same batch of GC analysis. 

24.   Line 208 – did the methods describe how the biofilm was grown without packing?  And how was total biofilm mass determined in the absence of packing?

 The detail method has been added in line #97-98# in section 2.3. All the biofilm has been collected by filter with Whatman filter paper grade 4 and dried in 105 oC for 24hours until the dried mass constant. 

25.   Line 241 – is the metabolic cross-feeding known?  Can you point to specific compounds within your dataset to indicate that these are the metabolites of interest?

 We did not make the metabolic cross-feeding. This statement was made based on previous reference and figure 3D, 3E (comparing the growth of each strain in single and co-culture).

26.   Figure 4A – please be more specific.  For instance, the two-way arrows between the 3 species are a bit misleading, as some metabolites go one-way only, while others may go both ways.  Is it possible to clarify this?

 Figure 4A has been modified accordingly.

27.   Fig 4B is great and would be helpful to have earlier in the manuscript when describing the approach.

 Thank you for your comment. This figure was also mentioned earlier in Material and Method part (L96).

28.   Line 307 – maximum biofilm at air-liquid interface sounds more like results.  Is this due to oxygen requirements?  To mention this new data in the final paragraph may not be the best approach.

 Yes, exactly. The maximum biofilm at the air/liquid interface show the oxygen requirement of filamentous fungi. This sentence has been moved to section 3.2.1 (L212-213).

29.   Line 317 – I think the cultivation device is a tool that will help to enable steps toward developing an improved understanding of the interactions.  Please be careful of the wording in this sentence.

 Thank you very much. The sentence has been modified accordingly (L341-342). 

30.   Line 318-319 indicates “management of spatial organization” – I don’t see this being accomplished in the current work.  While there was a substrate for surface-associated communities to form, this wasn’t quite spatial organization.

 Thank you very much for this finding. We do agree with you. The phrase has been modified (L343)

31.   It would be helpful to have a conclusions paragraph or section.  The final paragraph is partially conclusions but partially new information.  Conclusions should not introduce new information but simply summarize the work and its findings.

 The conclusion paragraph has been made separately from discussion (From L336-344).

32.   Are these 3 strains sufficient to ferment rice wine?  Do more strains need to be added?

 According to our previous study (DOI: 10.3389/fmicb.2018.00894), those three strains were considered as potential strain found in ferment starter and at the end of fermentation. This means that the involvement of these three strains during rice wine fermentation process is important.
